# Physiological Concentrations of *Cimicifuga racemosa* Extract Do Not Affect Expression of Genes Involved in Estrogen Biosynthesis and Action in Endometrial and Ovarian Cell Lines

**DOI:** 10.3390/biom12040545

**Published:** 2022-04-05

**Authors:** Maša Sinreih, Klara Gregorič, Kristina Gajser, Tea Lanišnik Rižner

**Affiliations:** Institute of Biochemistry and Molecular Genetics, Faculty of Medicine, University of Ljubljana, 1000 Ljubljana, Slovenia; masa.sinreih@mf.uni-lj.si (M.S.); klara.gregoric@gmail.com (K.G.); gajserkristina@gmail.com (K.G.)

**Keywords:** *Cimicifuga racemosa*, endometrial cancer, ovarian cancer, steroid transporters

## Abstract

In postmenopausal women, estrogen levels exclusively depend on local formation from the steroid precursors dehydroepiandrosterone sulfate and estrone sulfate (E1-S). Reduced estrogen levels are associated with menopausal symptoms. To mitigate these symptoms, more women nowadays choose medicine of natural origin, e.g., *Cimicifuga racemosa* (CR), instead of hormone replacement therapy, which is associated with an increased risk of breast cancer, stroke, and pulmonary embolism. Although CR treatment is considered safe, little is known about its effects on healthy endometrial and ovarian tissue and hormone-dependent malignancies, e.g., endometrial and ovarian cancers that arise during menopause. The aim of our study was to examine the effects of CR on the expression of genes encoding E1-S transporters and estrogen-related enzymes in control and cancerous endometrial and ovarian cell lines. CR affected the expression of genes encoding E1-S transporters and estrogen-related enzymes only at very high concentrations, whereas no changes were observed at physiological concentrations of CR. This suggests that CR does not exert estrogenic effects in endometrial and ovarian tissues and probably does not affect postmenopausal women’s risks of endometrial or ovarian cancer or the outcomes of endometrial and ovarian cancer patients.

## 1. Introduction

Around menopause, most women experience symptoms such as hot flashes, difficulty sleeping, and mood changes. To improve the quality of life of women with menopausal symptoms, hormone replacement therapy is routinely prescribed [1]. However, the correlation between hormone replacement therapy and an increased risk of breast cancer [2] and stroke [3] has led to the reduced use of such therapy [4]. As an alternative to hormone replacement therapy, plant-derived drugs, such as *Cimicifuga racemosa* (CR, black cohosh) extract, are used to alleviate menopausal symptoms. CR is a perennial dicot plant from the Ranunculaceae family, native to Canada and the Eastern United States, from which several different compounds have been isolated, including phenols, chromones, triterpenoids, and nitrogen-containing constituents [5]. CR extract has been used for centuries, and clinical studies have shown that CR reduces the occurrence of menopausal symptoms [6,7] and is safe, with no systemic [8,9] or breast-specific estrogenic effects [9,10]. Furthermore, no changes in endometrial thickness [10,11,12] or the occurrence of endometrial hyperplasia or cancer were observed [13]. However, the latest Cochrane report from 2012 based on data from 2027 peri- or postmenopausal women concluded that “there is currently insufficient evidence to support the use of black cohosh for menopausal symptoms, although there is adequate justification for conducting further studies in this area” [14].

In vivo studies on ovariectomized rats showed that CR extract prevents hot flashes [15] and exerts osteoprotective effects without estrogenic effects in the uterus [16] or mammary glands [17]. Reports also showed that CR extract does not exert estrogenic effects on the endometrial cancer (EC) cell line Ishikawa (as demonstrated by evaluating the effects on target genes and the induction of alkaline phosphatase [18]); the inhibition of the estradiol-induced proliferation of MCF-7 breast cancer cells [19]; growth promotion in MCF-7 or MDA-MB-231 breast cancer cells; or changes in estrogen levels [20]. Studies suggest that the mechanism of action of CR may be centrally mediated, with possible action at the level of serotonin or dopamine receptors [21], whereas CR does not bind to the estrogen receptors α and β [22]. CR extract inhibits the proliferation of the EC cell line Ishikawa [23] and breast cancer cells [24,25,26], suggesting that CR treatment may be a safe remedy for menopausal symptoms in breast cancer patients. However, the mechanism of action that inhibits proliferation is still unclear. To the best of our knowledge, only two studies have examined the effects of CR in premenopausal EC [18,23], whereas no studies have investigated the possible safety or effects of CR on estrogen formation or postmenopausal EC or ovarian cancer (OC).

EC and OC are very common gynaecological malignancies. EC arises from the lining of the uterus and is mostly diagnosed as a low-grade, early-stage disease. Most EC cases are diagnosed in postmenopausal patients and can be classified into two types. Type 1 is estrogen-dependent, in which development and progression occur due to endo- or exogenous estrogen exposure; this type is unopposed by progesterone or progestins and has a favorable prognosis. Type 2 presents approximately 10% of cases and is more aggressive, with a poorer prognosis. Integrated genomic characterisation has further stratified EC cases into four groups: the *POLE* ultramutated, microsatellite instability hypermutated, copy number low, and copy number high groups [27,28]. Recently, prognostic subgroups have been confirmed, enabling targeted therapy [29].

Among gynaecological cancers, OC is the leading cause of death in the developed world, with most cases diagnosed at stage III or IV of the disease. OC is a heterogeneous disease and can be categorised into four primary histological subtypes: serous, endometrioid, mucinous, and clear cell. High-grade serous cancers represent 70–80% of all OC cases [30,31]. Higher endogenous estrogen exposure through life increases the risk of OC [31]. Recently, high-grade serous OC has been divided into four molecular subtypes with distinct genomic profiles: mesenchymal, differentiated, immunoreactive, and proliferative [32].

In postmenopausal women, active estrogens can still be synthesized from inactive precursors, such as adrenal dehydroepiandrosterone and dehydroepiandrosterone sulfate, ovarian androstenedione, or circulating estrone sulfate (E1-S). To enable E1-S metabolism towards estradiol (E2), E1-S uptake must occur. Transporters that promote E1-S uptake are encoded by the solute carrier gene super family (SLC), more precisely by the SLC21, SLC22, and SLC10 subfamilies. Most transporters from the organic anion transporting polypeptides (OATP) that are encoded by the genes from the solute carrier organic anion (SLCO) SLC21 subfamily can catalyse E1-S uptake. In addition, some organic anion transporters, encoded by genes from the SLC22 subfamily, are also capable of E1-S uptake. The SLC10A6 transporter from the sodium-dependent organic anion transporter SLC10 subfamily is an uptake transporter for steroids as well [33].

E1-S efflux transporting pumps belong to the ATP-binding cassette (ABC) transporter family, from which ABCC1, ABCC4, and ABCG2 mediate E1-S transport. Additionally, organic solute transporters α and β (heterodimers encoded by *SLC51A* and *SLC51B* from the SLC51 gene family) can promote E1-S efflux. However, because they mediate transport along the electrochemical gradient, they can also promote E1-S uptake [34,35].

Metabolism towards the most active estrogen, E2, occurs in different peripheral tissues but preferentially in adipose tissue, where it can be synthesized either by the aromatase pathway from dehydroepiandrosterone sulfate and dehydroepiandrosterone or by the sulfatase pathway from E1-S. We recently showed that in menopausal EC patients, E2 formation more likely results from the sulfatase pathway, which is also the case in the adjacent control endometrium, with increased levels of E2 seen in cancerous tissues [36]. Here, steroid sulfatase (STS) and sulfotransferase 1E1 (SULT1E1) direct the formation of E1-S or estrone (E1), and 17β-hydroxysteroid dehydrogenase 1 (HSD17B1) and 2 (HSD17B2) direct the formation of E2 or E1 (Figure 1). The active forms of estrogens can promote cell proliferation, for which estrogen receptors α and β play a crucial role. Active estrogens promote rapid cell multiplication, and thus errors in DNA sequences can appear, consequently causing carcinogenesis [34,35].

To date, the effects of CR are still poorly understood, including its effects on steroid precursor import, estradiol synthesis, estrogen metabolism, metabolite elimination, and active estrogen concentrations in hormonally dependent endometrial and ovary tissues. The aim of this study was to elucidate the mechanism of action and possible protective or stimulating effects of CR extracts on EC and OC development.

## 2. Materials and Methods

### 2.1. Cimicifuga Racemosa

Dry extract of CR rhizome (BNO 1055, batch No. 770119, Bionorica, Nuemartkt in der Oberpfalz, Germany) was dissolved and diluted in 50% (*v*/*v*) ethanol before application to cell cultures.

### 2.2. Cell Culture

The HEC-1-A (CVCL_0293) cell line was originally established from moderately differentiated endometrial adenocarcinoma from a 71-year-old patient and was purchased from the American Type Culture Collection (ATCC_HTB-112^TM^) as p125 on 31 May 2012. McCoy’s 5A Medium (M4892, Sigma-Aldrich St. Louis, MO, USA) supplemented with 10% foetal bovine serum (FBS) was used as growth medium. McCoy’s 5A Medium without phenol red (SH30270.01, GE Healthcare Life Sciences, Piscataway, NJ, USA) was used as treatment medium. HEC-1-A passage 15 (p+15) cells were authenticated by short tandem repeats (STR) profiling performed by ATCC on 22 February 2018.

The Ishikawa (CVCL_2529) cell line was originally established from a well differentiated endometrial adenocarcinoma from a 39-year-old patient and was purchased from Sigma-Aldrich (ECACC 99040201) as p+3 on 18 December 2012. Minimum Essential Medium Eagle (#M5650) with 2 mM L-glutamine (#G7513) and 5% FBS (#F9665, Sigma-Aldrich St. Louis, MO, USA) was used as growth medium. MEM without phenol red (#51200-038, Thermo Fisher Scientific, Waltham, MA, USA) and supplemented with 2 mM L-glutamine (#G7153) was used as treatment medium. Ishikawa p+13 cells were authenticated by STR profiling performed by ATCC on 22 February 2018.

The RL95-2 (CVCL_0505) cell line was originally established from a Grade 2 moderately differentiated endometrial adenosquamous carcinoma from a 65-year-old patient [37] and was purchased from the American Type Culture Collection (ATCC_CRL-1671™, lot 62130010) as p125 on 18 October 2017. DMEM:F12 (D6421) with 10% FBS (F9665), 2.5 mM L-glutamine (G7153), and 5 µg/mL insulin (I9278; all from Sigma-Aldrich St. Louis, MO, USA) was used as growth medium. DMEM:F12 without phenol red (D6434) and supplemented with 2.5 mM L-glutamine (G7153) and 5 µg/mL insulin (I9278; all Sigma-Aldrich St. Louis, MO, USA ) was used as treatment medium.

The KLE (CVCL_1329) cell line was originally established from a poorly differentiated endometrial carcinoma from a 68-year-old patient (Richardson et al., 1984) and was purchased from the American Type Culture Collection (ATCC_CRL-1622™, lot 70001143) as p+12 on 18 October 2017. DMEM:F12 (D6421) supplemented with 10% FBS (F9665) and 2.5 mM L-glutamine (G7153; all from Sigma-Aldrich St. Louis, MO, USA) was used as growth medium. DMEM:F12 without phenol red (D6434) and supplemented with 2.5 mM L-glutamine (G7153; all from Sigma-Aldrich GmbH) was used as treatment medium.

The control cell line HIEEC was obtained from Michael A. Fortier (Laval University, Quebec, QC, Canada) as p14 on 4 April 2014. It was originally generated from a primary culture prepared from an endometrial biopsy from a 37-year-old woman with confirmed absence of neoplasia and endometriosis, on day 12 of her menstrual cycle [38]. RPMI-1640 Medium (R5886) supplemented with 2 mM L-glutamine (G7153) and 10% FBS (F9665; all from Sigma-Aldrich St. Louis, MO, USA) was used as growth medium. RPMI-1640 medium without phenol red (11835-030, Thermo Fisher Scientific, Waltham, MA, USA) was used as treatment medium. HIEEC p+8 cells were authenticated by STR profiling performed by ATCC on March 8, 2018. 

The Kuramochi (CVCL_1345) cell line was originally established from high-grade ovarian serous adenocarcinoma from a metastatic site in the ascites (Motoyama, 1981) and was purchased from JCRB (JCRB0098 lot 06302015) as p17 on 23 October 2017. RPMI (R5886) with 10% FBS (F9665) and 2 mM L-glutamine (G7153; all from Sigma-Aldrich St. Louis, MO, USA) was used as growth medium. RPMI without phenol red (11835-030; Thermo Fisher Scientific, Waltham, MA, USA) was used as treatment medium. 

The COV362 (CVCL_2420) cell line was originally established from a high-grade ovarian serous adenocarcinoma derived from a metastatic site in pleural effusion (van den Berg-Bakker et al., 1993) and was purchased from ECACC (07071910) as p37 on 13 October 2017. DMEM (D5546) with 10% FBS (F9665) and 2 mM L-glutamine (G7153; all from Sigma-Aldrich GmbH) was used as growth medium. DMEM without phenol red (D5921) and supplemented with 2 mM L-glutamine (G7153; all from Sigma-Aldrich St. Louis, MO, USA) was used as treatment medium. 

The OVSAHO (CVCL_3144) cell line was originally established from a serous papillary adenocarcinoma from a metastatic site in the abdomen (Yanagibashi et al., 1997) of a 56-year-old woman, representing a suitable model of high-grade serous OC, and was purchased from JCRB (JCRB1046 lot 04062015) as p44 on 4 June 2018. RPMI (R5886) with 10% FBS (F9665) and 2 mM L-glutamine (G7153; all from Sigma-Aldrich, St. Louis, MO, USA) was used as growth medium. RPMI without phenol red (11835-030; Thermo Fisher Scientific, Waltham, MA, USA) was used as treatment medium. 

The control cell line HIO80 (CVCL_E274) was originally established from ovarian surface epithelium (Yang et al., 2004) and was obtained from Andrew K. Godwin (University of Kansas Medical Center, USA) as p+72 on 20 October 2017. RPMI (#R5886) supplemented with 10% FBS (F9665), 2 mM L-glutamine (G7153), and 5.6 μg/mL insulin (I9278; all from Sigma-Aldrich St. Louis, MO, USA) was used as growth medium. RPMI without phenol red (11835-030; Thermo Fisher Scientific, Waltham, MA, USA) and supplemented with 5.6 μg/mL insulin (I9278) was used as treatment medium. The cells were authenticated by STR profiling performed by ATCC on 25 February 2019.

All cell lines were negative for mycoplasma infection, which was periodically tested with the MycoAlert^TM^ mycoplasma detection kit (Lonza, Basel, Switzerland). STR profiling was performed by ATCC on cell lines that were purchased from culture collections several years prior or obtained from other laboratories.

### 2.3. Treatment of Cell Lines with CR Extract

Cells from different cell lines were seeded onto 6-well plates (3.00 × 10^5^ KLE cells/well, 3.50 × 10^6^ RL95-2 cells/well, 5.00 × 10^5^ Kuramochi cells/well, 3.50 × 10^5^ COV362 cells/well, 1.10 × 10^5^ HIO80 cells/well, 1.50 × 10^5^ HIEEC cells/well, 1.50 × 10^6^ OVSAHO cells/well, 6.00 × 10^5^ HEC-1-A cells/well, and 3.00 × 10^5^ Ishikawa cells/well) in 2 mL of respective complete growth medium and reached 70% confluence within 24 h after seeding. One day after seeding, the medium was aspirated, and cells were washed with 2 mL of PBS. Next, 2 mL of respective treatment medium and 2 μL of CR extract (at final concentrations of 5 ng/mL, 500 ng/mL, 25 μg/mL, 50 μg/mL, and 100 μg/mL; dissolved in 50% ethanol) were added (Appendix A). For controls, we added only 2 μL 50% ethanol to the treatment media. Cells grew in treatment medium for 72 h. After 72 h, the medium was aspirated, cells were washed with PBS, and RA1 lysis buffer was added, according to the manufacturer’s instructions (Macherey-Nagel GmbH & Co. KG, Düren, Germany). Experiments were performed in three independent replicates.

### 2.4. RNA Isolation and Quantitative Real-Time PCR

Total RNA from cells was isolated and purified after cultivation in treatment medium using Nucleospin RNA isolation kits (Macherey-Nagel GmbH & Co. KG, Düren, Germany), according to the manufacturer’s instructions. The quality of RNA was determined using the Agilent 2100 Bioanalyzer and RNA 600 Nanokit (Agilent Technologies Inc., Santa Clara, CA, USA). The measured RNA integrity number values were above 9.0, indicating that the RNA was of good quality. Total RNA was reverse transcribed into cDNA using the SuperScript^®^ VILO™ cDNA Synthesis kit (Invitrogen, Thermo Fisher Scientific, Carlsbad, CA, USA) according to manufacturer’s instructions. The cDNA samples were stored at −20 °C.

The expressions of genes that encode estrogen receptors and proteins involved in estradiol biosynthesis and oxidative metabolism were examined using quantitative PCR (qPCR). The following was used: exon-spanning hydrolysis probes commercially available as ‘Assay on Demand’ (Applied Biosystems; Foster City, CA, USA) (Table 1), using TaqMan^®^ Fast Advanced Master Mix. The expressions of genes that encode for transporters were examined using SYBR Green I Master (Roche, Basel, Switzerland) and primers that were designed in our laboratory (Table 2). Quantification was accomplished with the Applied Biosystems^®^ ViiA™ 7 Real-Time PCR System (Thermo Fisher Scientific, Waltham, MA, USA). All the cDNA samples were run in triplicates, using 0.25 μL of cDNA, and the reactions were performed in Applied Biosystems^®^ MicroAmp^®^ Optical 384-well plates (Thermo Fisher Scientific, Waltham, MA, USA) in a reaction volume of 5 μL. For gene expression analysis, the normalization factor for each sample was calculated based on the geometric mean of the three most stably expressed reference genes (*POLR2A*, *HPRT1*, *RPLP0*). Gene expression for each sample was calculated from the crossing-point value (Cq) as *E*^−Cq^, divided by the normalization factor and multiplied by 10^12^. The Minimum Information for Publication of Quantitative Real-Time PCR Experiments guidelines were considered in the performance and interpretation of the qPCR reactions [39].

### 2.5. Median Cytotoxic Concentration (CC50)

After the cells were grown in treatment medium for 72 h, the medium was aspirated, and the cells were washed with PBS and detached with trypsin. Detached cells were resuspended in 1 mL of growth medium. Cell suspensions were mixed with trypan blue dye (1:1) and used for cell counting with an automated cell counter. Cell viability was calculated as a ratio between live treated (with CR extract dissolved in 50% ethanol, final concentration of ethanol was 0.5%) and control cells (only 50% ethanol). Experiments were performed in at least two independent replicates.

### 2.6. xCELLigence

For real-time proliferation monitoring using the xCELLigence RTCA DP system (Agilent, Santa Clara, CA, USA), KLE cells were seeded onto E-plates 16 (ACEA Biosciences, San Diego, CA, USA) at cell densities of 5000 cells/well in growth medium without phenol red and with charcoal-stripped FBS (#F6765, Sigma-Aldrich). The next day, KLE cells were treated with different concentrations of CR extract (20–400 μg/mL; Appendix A), only 50% ethanol, or only medium. The final concentration of ethanol was 2.5%. Experiments were repeated in two independent experiments, each time with four technical replicates for individual treatments. Cell proliferation was monitored for 180 h after treatment.

### 2.7. Statistical Analysis

Statistical analysis was performed in Graph Pad Prism 8.0.0 for Windows (GraphPad Software, San Diego, CA, USA). Values of gene expression in individual cell lines for treated and untreated (control) samples was normalized with the expression of reference genes. Data were statistically analysed via the Kruskal–Wallis test with Dunn’s multiple comparisons test. *p* < 0.05 was considered statistically significant.

## 3. Results and Discussion

### 3.1. CC50 Values of CR Were Higher in EC and OC Cell Lines Compared to Those in Control Cell Lines


We first examined the CC_50_ values of the CR extract in all nine cell lines (Appendix A). Cell viability decreased with higher CR concentrations in a dose-dependent manner. In EC cell lines, CC_50_ values were in the range of 20.16–58.23 μg/mL (Figure 2, Table 3. The lowest values among EC cell lines were measured in RL-95-2 cells. The control cell line HIEEC had an even lower CC_50_ value: 11.82 μg/mL. In OC, the highest CC_50_ was determined in OVSAHO cells (106.60 μg/mL), whereas Kuramochi and COV362 cells had lower CC_50_ values: 63.91 and 63.48 μg/mL, respectively. Similarly to EC cell lines, the control cell line HIO80 had lower CC_50_ values for CR extract (12.56 μg/mL) than those of cancerous cells. Our results show that cancer cells can survive at much higher concentrations of CR extract than control cells. Hostanska et al. determined 50% growth inhibitory concentrations for ethanol extracts in breast cancer cell lines, which are in the same range as our results (12.6–29.5 μg/mL) [24].

For KLE cells, which represent a model of poorly differentiated EC with bad prognosis, cell proliferation after CR treatment was measured in real time. Cell death occurred with 400 or 300 μg/mL of CR extract, whereas lower CR concentrations only reduced the growth rate, while 100% confluence could still be reached (Figure 3).

### 3.2. At Physiological Concentrations, CR Extract Did Not Alter the Expression of the Investigated Genes in Control and EC Cell Lines

Cells were treated with concentrations similar to the plasma concentrations of the 23-epi-26-deoxyactein, a major constituent of CR, measured in women using CR supplements (5–10 ng/mL) [40] and with higher concentrations (up to 100 µg/mL, depending on the cell line, Appendix A). Next, we examined the expression of 10 different genes involved in uptake (*SLCO1A2*, *SLC10A6*, *SLCO2B1*, *SLCO3A1*, *SLCO4A1*, *SLCO4C1*, *SLCO1B1*, *SLCO1B3*, *SLCO1C1*, *SLC22A11*); five genes involved in efflux (*ABCC1, ABCC4, ABCG2, SLC51A, SLC51B*); and six genes involved in estradiol biosynthesis, metabolism, and action (*ESR1*, *ESR2*, *HSD17B1*, *HSD17B2*, *STS*, *SULT1E1*).

When the control endometrial cell line HIEEC was treated with CR extract (5 and 500 ng/mL), no changes in the expression of the investigated genes were detected (Figure 4, Figure 5 and Figure 6). The Ishikawa and HEC-1-A cell lines represent well-differentiated EC and were established from a premenopausal and postmenopausal EC patient, respectively. Physiological CR concentrations exhibited no effects on gene expression. Only at 500 ng/mL, Ishikawa cells exhibited downregulated expression of *SLCO1A2*, which encodes an influx transporter. RL-95-2 cells are derived from Grade 2 moderately differentiated EC. Similarly, CR did not affect gene expression, except for *ESR1,* which was downregulated at very high CR concentrations (100 μg/mL). Our results suggest that CR does not influence estrogen concentration and actions in either normal endometrial cells or well- to moderately differentiated endometrial cancer.

### 3.3. High Concentrations of CR Extract Greatly Affected the Expression of Estrogen-Related Genes in KLE Cells

KLE cells are derived from poorly differentiated endometrial carcinoma Grade 3 and were the most affected by CR extract, which influenced both genes encoding influx and efflux transporters and genes involved in estradiol biosynthesis, metabolism, and action. At 100 μg/mL of CR extract, one gene encoding influx transporters (*SLCO2A1*) was downregulated, whereas two (*SLC10A6* and *SLCO4A1*) were upregulated. Similarly, three of the five examined efflux transporters (*ABCC1*, *ABCC4*, and *ABCG2*) were upregulated at higher CR concentrations. *ESR1* was upregulated at both 50 and 100 μg/mL of CR, and *STS* was downregulated at the highest CR concentration (Figure 7). This suggests a higher flux of estrogens through cells, but lower concentrations of the active estrogen estradiol. Transporters are not specific for E1-S transport, and their differential expression can also affect other processes such as ion transport, signal transduction, and toxin secretion.

Although lower CR concentrations did not affect control or cancer cell lines, our results show that the poorer the differentiation, the more affected the expressions of the investigated genes were. The concentrations that affected gene expression were mostly higher than the CC_50_ values for individual cell lines, except for *SLCO2A1* downregulation in Ishikawa cells and *ABCG2* upregulation in KLE cells. 

In EC, estrogens promote proliferation via estrogen receptor α, which is encoded by the *ESR1* gene. With cancer progression, *ESR1* levels drop, the ratio between ERα and ERβ shifts, and the loss of ERα is associated with shorter disease-free survival [41,42]. Thus, higher ESR1 levels would be beneficial, especially in poorly differentiated cancers. 

### 3.4. Higher Concentrations of CR Extract Upregulated the Expression of Influx and Efflux Transporter Genes in Ovarian Control Cell Line HIO-80

HI0-80, a control cell line established from the ovarian surface, was treated with CR extract (5 ng/mL, 500 ng/mL, 25 μg/mL, and 50 μg/mL). Physiological CR concentrations (5 ng/mL) did not affect gene expression. The concentrations that affected gene expression were four-fold higher than the CC_50_ values for HIO-80 cells. While the expression of most genes encoding influx and efflux transporters was higher at 25 and 50 μg/mL of CR extract, statistically significant changes were demonstrated for *SLCO2B1*, *SLC22A11*, and *SLC51B*. At 50 μg/mL of CR, *SULT1E1* expression was significantly downregulated (Figure 8). This suggests the higher flux of estrogens into the cells. If mRNA levels of *SULT1E1* correlate to protein levels, less estrone and estradiol is sulphated and higher levels of active estrogens could promote the proliferation of cells.

### 3.5. Higher CR Concentrations Affected the Expression of Influx Transporter Genes and Genes Involved in Estradiol Biosynthesis, Metabolism, and Action in the High-Grade Serous OC Cell Lines Kuramochi and COV362

We next studied the effects of CR in the OC cell lines Kuramochi, COV362, and OVSAHO. Although all three cancer cell lines were established from high-grade serous OC, they differ in aggressiveness and chemoresistance. In 2016, Haley et al. compared the ability of high-grade serous OC cell lines to migrate, invade, proliferate, and form colonies. OVSAHO cells exhibited the lowest functional activity, migration time, invasion time, and colony formation time. COV362 and Kuramochi cells did not differ in migration or invasion time, whereas colony formation time was shorter in Kuramochi cells, indicating a more aggressive type of cancer cell line. When chemoresistance was examined, COV362 cells were the most resistant against platinum-based drugs and had the highest IC_50_ value, whereas OVSAHO cells had the lowest IC_50_ value [43].

Similar to EC cancer cell lines and the control cell line HIO-80, no gene expression changes were observed at physiological CR concentrations. The most abundant genes encoding influx transporters in all three OC cell lines were *SLCO3A1*, *SLCO4A1*, and *SLCO4C1* (Figure 9), whereas the most highly expressed efflux transporter genes were *ABCC1* and *ABCC4* (Figure 10). In Kuramochi cells, *SLCO4A1* and *SLCO1B3* were downregulated at 100 μg/mL and 50 μg/mL of CR, respectively, whereas *SLCO4C1* was upregulated at 500 ng/mL of CR. In COV362 cells, *SLCO2B1* levels were lower at 100 μg/mL of CR. No changes were observed regarding influx transporters in OVSAHO cells. Out of all efflux transporter genes, only *ABCC1* levels were higher in COV362 cells.

Among genes involved in estradiol biosynthesis, metabolism, and action, *ESR1* was downregulated in COV362 cells at 50 and 100 μg/mL of CR, and *SULT1E1* was downregulated at 50 μg/mL of CR. In Kuramochi cells, *STS* was downregulated at 100 μg/mL of CR (Figure 11). The concentrations that affected gene expression were close to or higher than the CC_50_ value for the cell line, except for *SLCO4C1* in Kuramochi cells. 

Overall, if protein levels correlate with mRNA levels, Kuramochi cells would have lower E1-S influx and less E2 formed in the cell after CR treatment, COV362 cells would exhibit reduced estrogen action due to decreased *ESR1* levels, and OVSAHO cells would exhibit unaltered estrogenic properties.

To date, to the best of our knowledge, no studies have explored the effects of CR on genes involved in estrogen action in endometrial and ovarian cell lines; however, such a study has been carried out in the breast cancer cell line MCF-7. Gaube et al. showed that 15 μg/mL of lipophilic CR extract affected the expression of antiproliferative and proapoptotic genes, including downregulating *ESR1*; the authors concluded that this could contribute to the antitumor activity of CR [44].

## 4. Conclusions

Our research presents insight into the effects of CR on model cell lines of normal endometrial and ovarian tissue and EC and OC. Our results reveal that CR affects the expression of genes encoding E1-S transporters and estrogen-related enzymes only at very high concentrations (50 and 100 μg/mL). However, no changes in the expression of genes involved in E1-S influx or efflux, estrogen synthesis, or estrogen receptors were observed with concentrations similar to those detected in the plasma of CR users. These findings support previously published studies that demonstrated that CR extract most likely does not exert estrogenic effects or affect postmenopausal women’s risks of EC and OC or the outcomes of EC and OC patients.

## Figures and Tables

**Figure 1 biomolecules-12-00545-f001:**
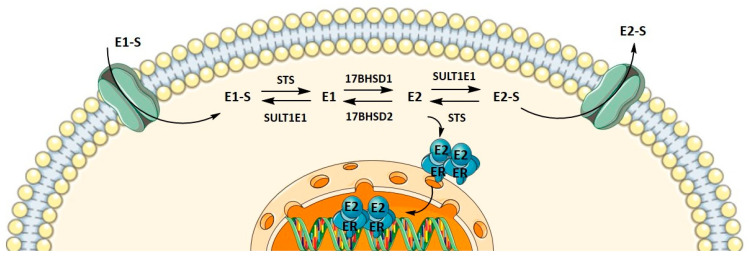
Uptake, intracrine action, and efflux of estrogens.

**Figure 2 biomolecules-12-00545-f002:**
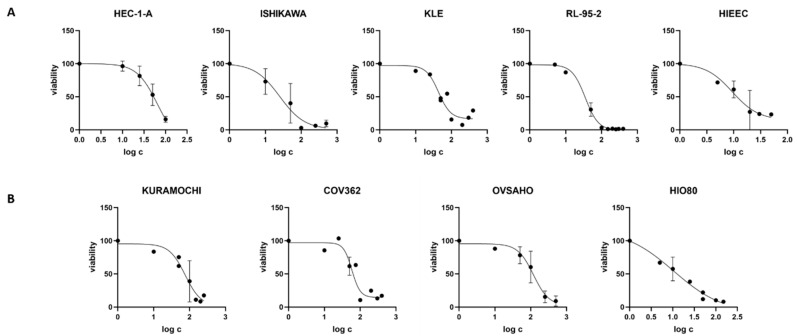
Cell viability after treatment with *Cimicifuga racemosa* extract. (**A**) The endometrial cancer cell lines HEC-1-A, Ishikawa, KLE, and RL-95-2 and the control cell line HIEEC. (**B**) The ovarian cancer cell lines Kuramochi, COV362, and OVSAHO and the control cell line HIO80. The mean values of at least two independent experiments with standard deviations are shown.

**Figure 3 biomolecules-12-00545-f003:**
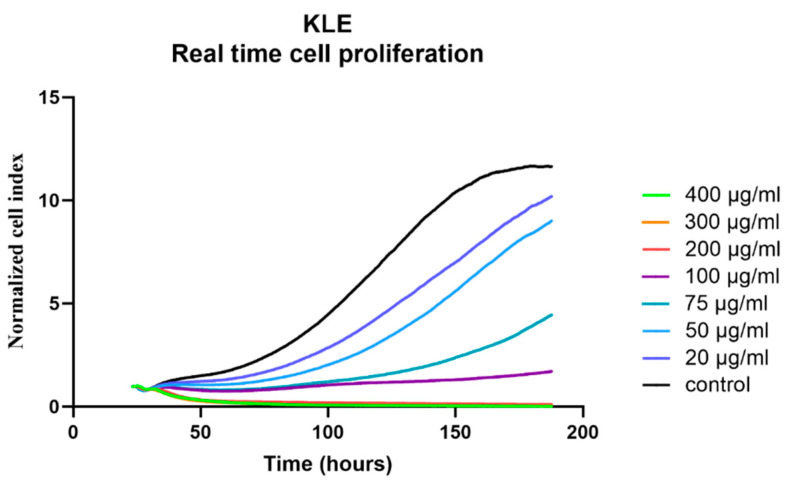
Real-time evaluation of the effects of *Cimicifuga racemosa* on KLE cell proliferation. Mean values of two independent experiments are shown.

**Figure 4 biomolecules-12-00545-f004:**
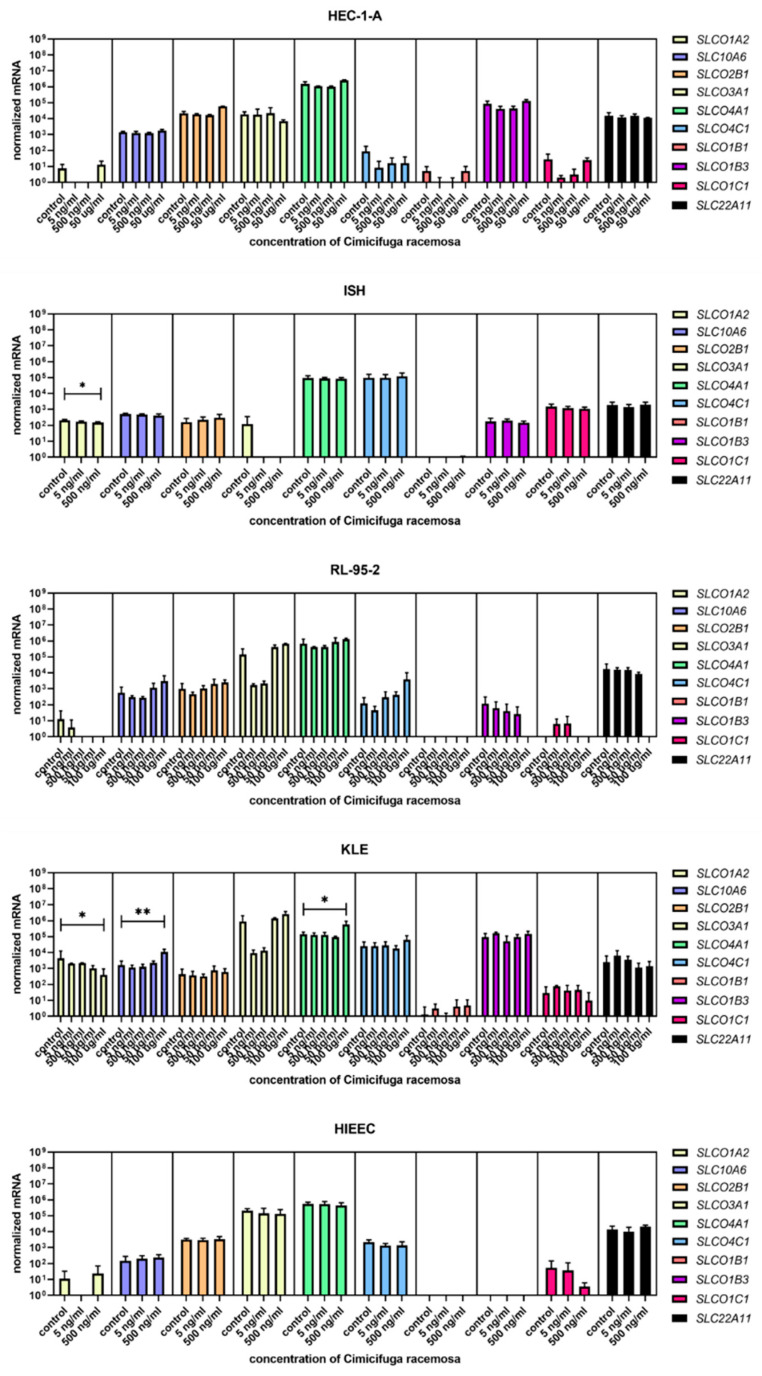
Normalized expression of 10 different genes of uptake (*SLC01A2*, *SLC10A6*, *SLCO2B1*, *SLCO3A1*, *SLCO4A1*, *SLCO4C1*, *SLCO1B1*, *SLCO1B3*, *SLCO1C1*, *SLC22A11*) in endometrial cancer cell lines HEC-1-A, Ishikawa, RL-95-2, and KLE and control cell line HIEEC. At least three independent experiments were performed. Data are shown as mean ± SD. * *p*-value ≤ 0.05, ** *p*-value ≤ 0.01.

**Figure 5 biomolecules-12-00545-f005:**
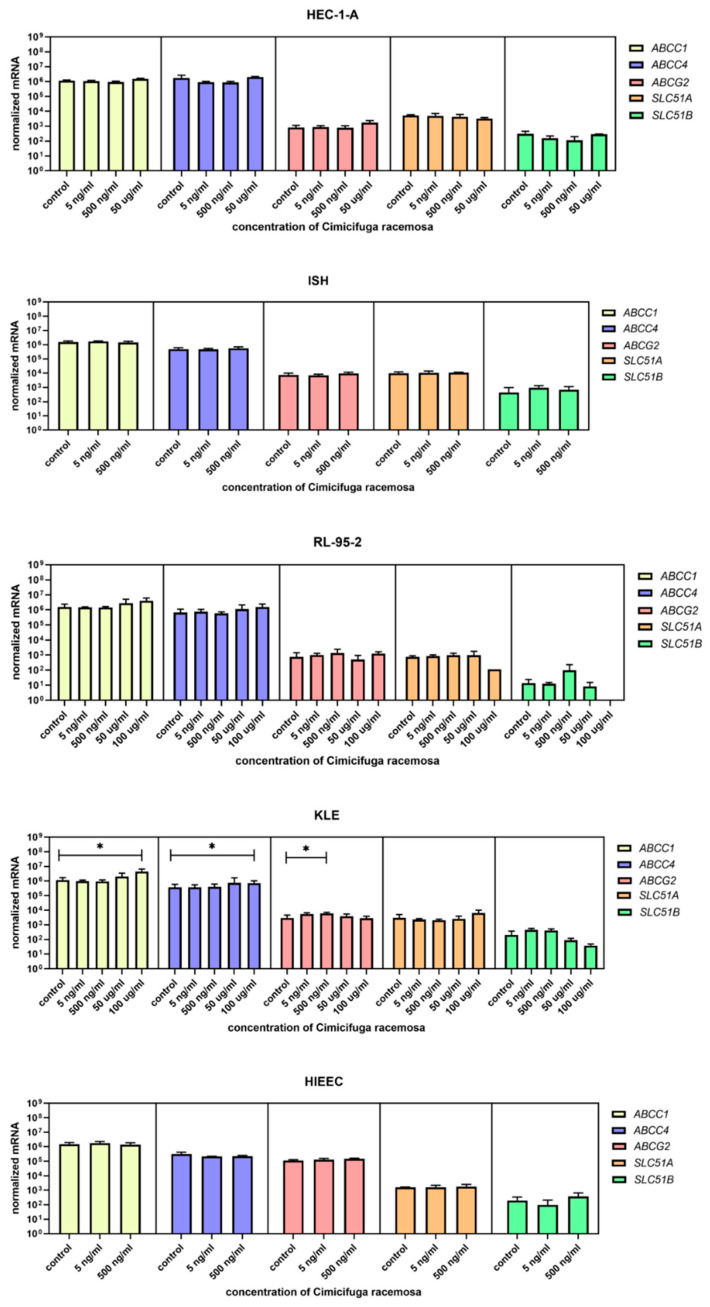
Normalized expression of five different genes of efflux (*ABCC1*, *ABCC4*, *ABCG2*, *SLC51A*, *SLC51B*) in endometrial cancer cell lines HEC-1-A, Ishikawa, RL-95-2, and KLE and control cell line HIEEC. At least three independent experiments were performed. Data are shown as mean ± SD. * *p*-value ≤ 0.05.

**Figure 6 biomolecules-12-00545-f006:**
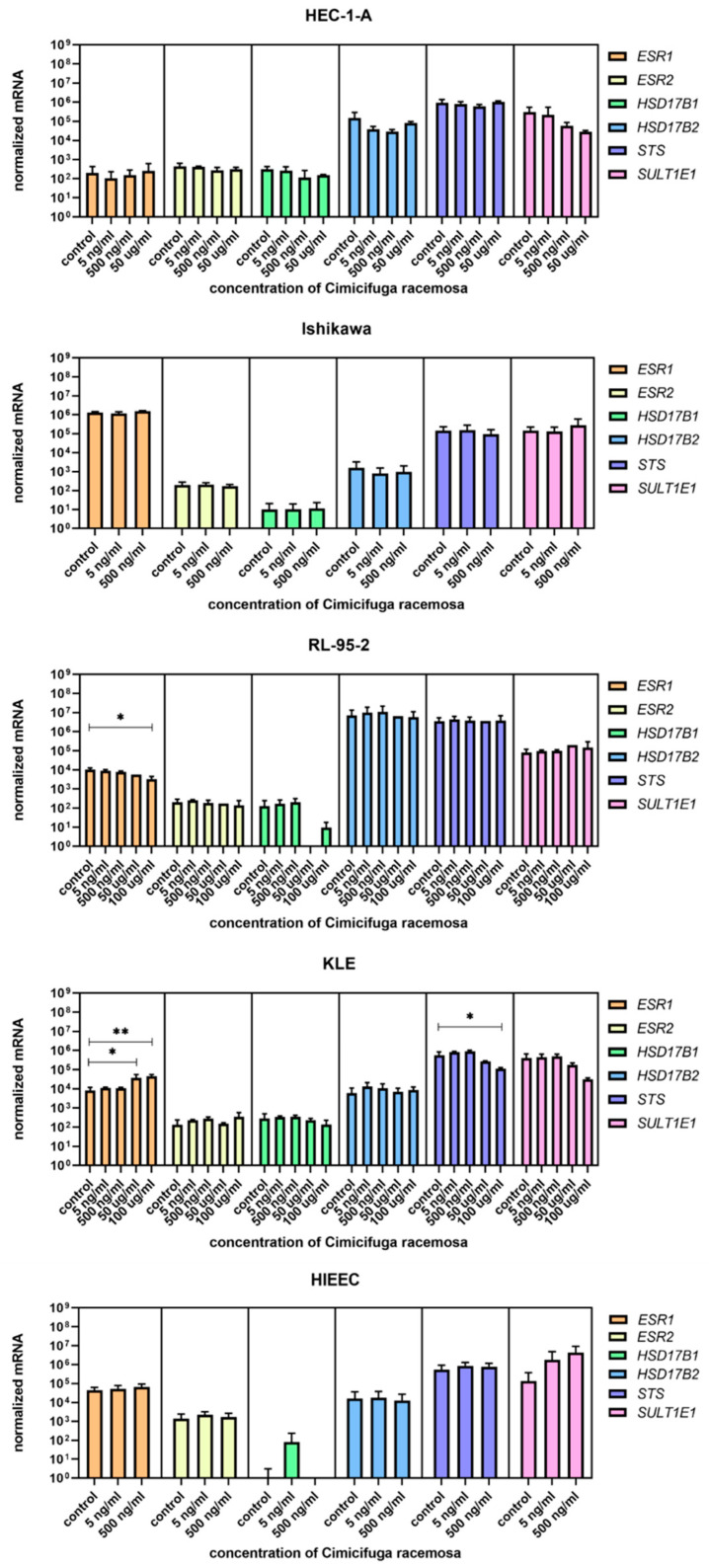
Normalized expression of genes encoding estrogen receptors and genes involved in estradiol biosynthesis and metabolism in endometrial cancer cell lines HEC-1-A, Ishikawa, RL-95-2, and KLE and control cell line HIEEC. At least three independent experiments were performed. Data are shown as mean ± SD. * *p*-value ≤ 0.05, ** *p*-value ≤ 0.01.

**Figure 7 biomolecules-12-00545-f007:**
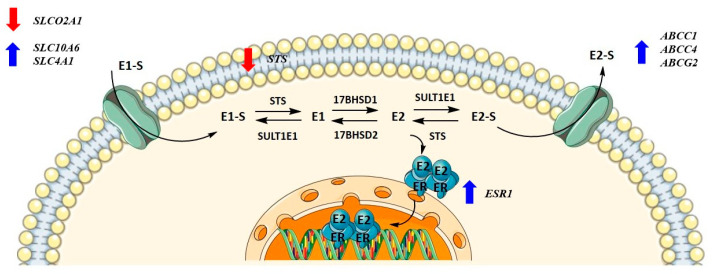
Changes in gene expression levels due to high concentrations of *Cimicifuga racemosa* extract in the KLE cell line (derived from poorly differentiated endometrial cancer).

**Figure 8 biomolecules-12-00545-f008:**
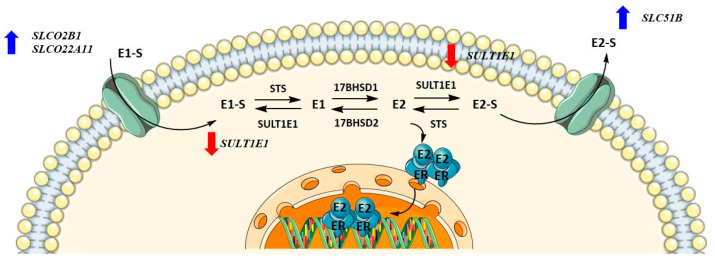
Changes in gene expression levels due to high concentrations of *Cimicifuga racemosa* extract in the HIO-80 cell line (control cell line of ovarian epithelium).

**Figure 9 biomolecules-12-00545-f009:**
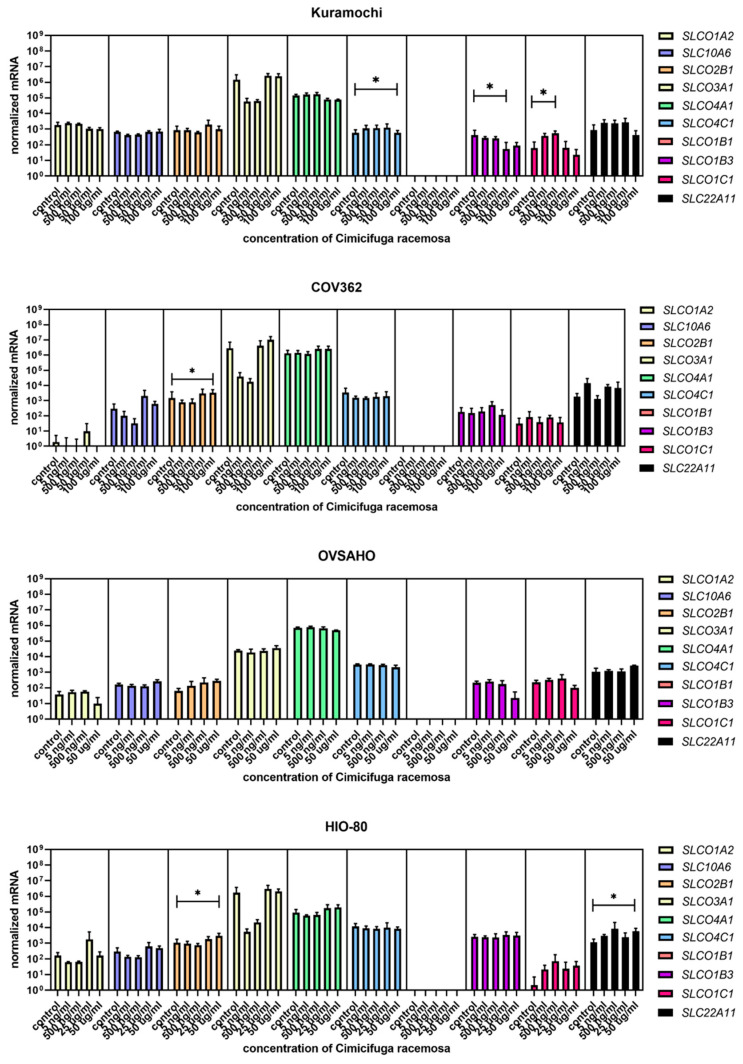
Normalized expression of 10 different genes of uptake (*SLC01A2*, *SLC10A6*, *SLCO2B1*, *SLCO3A1*, *SLCO4A1*, *SLCO4C1*, *SLCO1B1*, *SLCO1B3*, *SLCO1C1*, *SLC22A11*) in the ovarian cancer cell lines Kuramochi, COV362, and OVSAHO and control cell line HIO-80. At least three independent experiments were performed. Data are shown as mean ± SD. * *p*-value ≤ 0.05.

**Figure 10 biomolecules-12-00545-f010:**
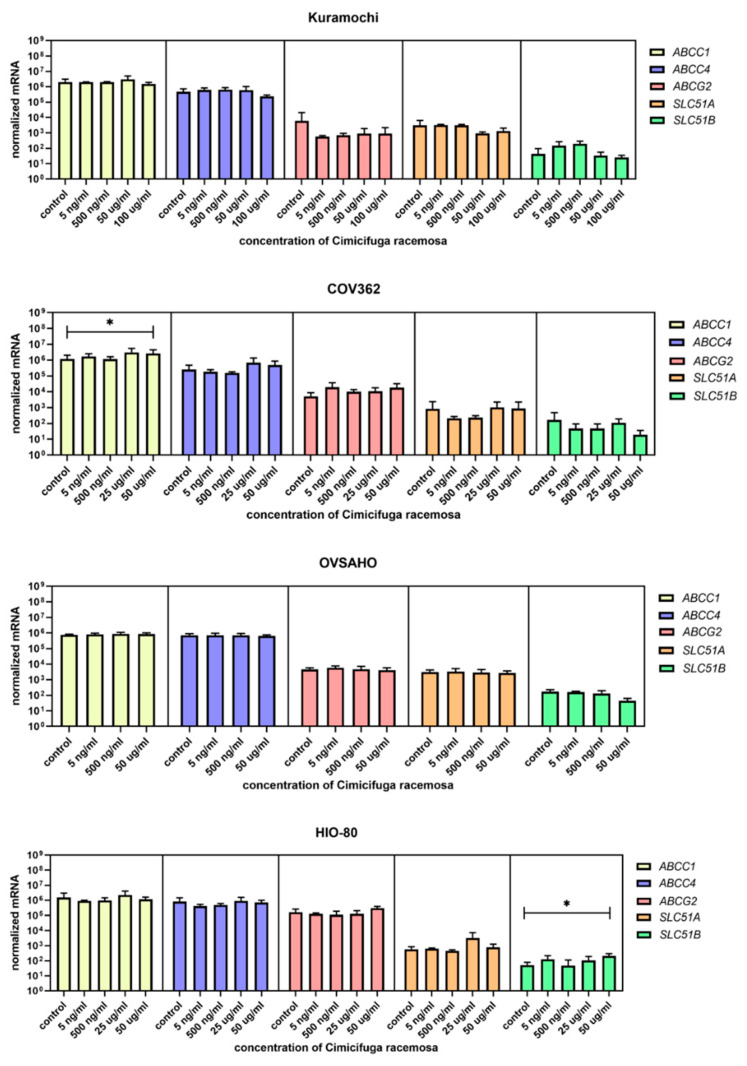
Normalized expression of five different genes of efflux (*ABCC1*, *ABCC4*, *ABCG2*, *SLC51A*, *SLC51B*) in the ovarian cancer cell lines Kuramochi, COV362, and OVSAHO and control cell line HIO-80. At least three independent experiments were performed. Data are shown as mean ± SD. * *p*-value ≤ 0.05.

**Figure 11 biomolecules-12-00545-f011:**
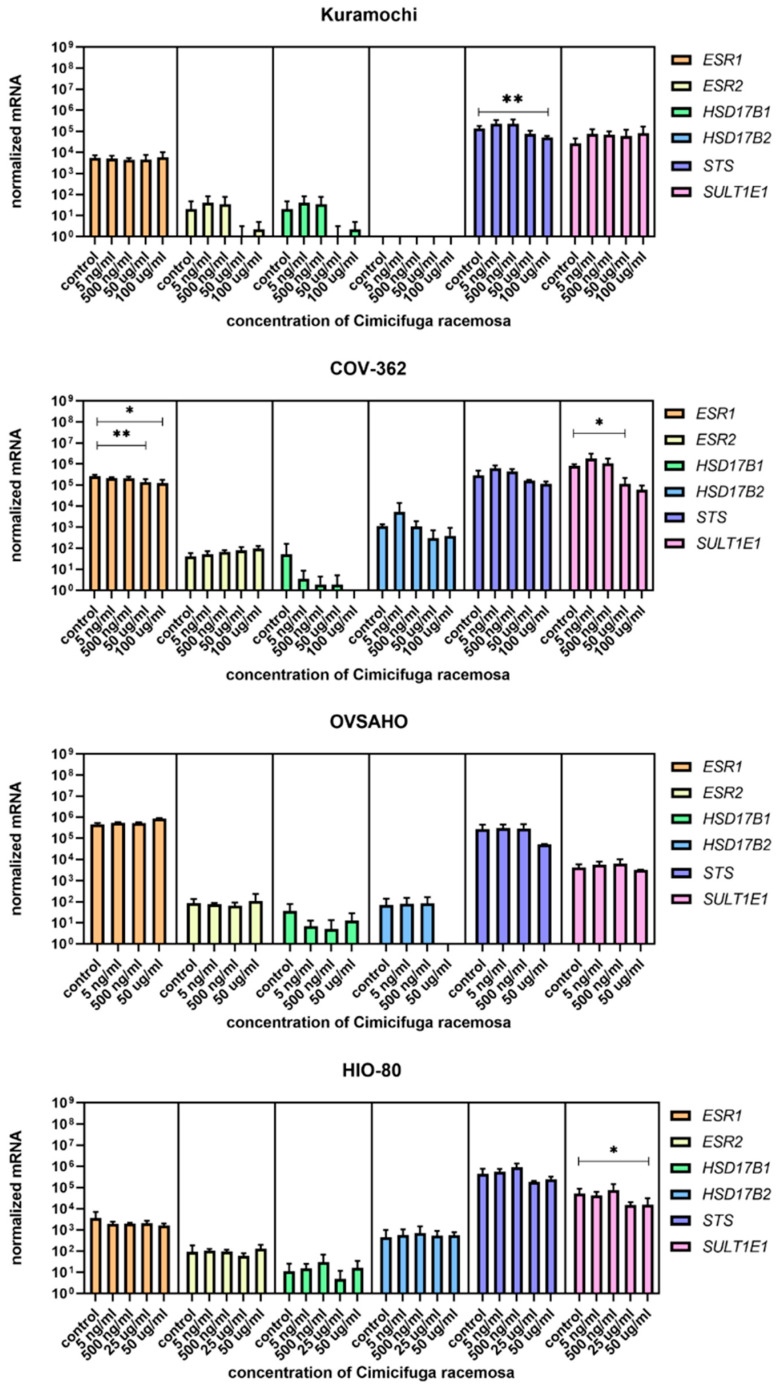
Normalized expression of genes encoding estrogen receptors and genes involved in estradiol biosynthesis and metabolism and genes encoding estrogen receptors in ovarian cancer cell lines Kuramochi, COV-362, and OVSAHO and control cell line HI0-80. At least three independent experiments were performed. Data are shown as mean ± SD. * *p*-value ≤ 0.05. ** *p*-value ≤ 0.01.

**Table 1 biomolecules-12-00545-t001:** TaqMan^®^ assays on demand for the genes investigated.

Gene Symbol	Assay ID	Gene Name
*ESR1*	Hs00174860_m1	Estrogen receptor 1
*ESR2*	Hs00230957_m1	Estrogen receptor 2 (ERβ)
*HPRT1* *	Hs99999909_m1	Hypoxanthine phosphoribosyltransferase 1 (Lesch-Nyhan syndrome)
*HSD17B1*	Hs00166219_g1	Hydroxysteroid (17β) dehydrogenase 1
*HSD17B2*	Hs00157993_m1	Hydroxysteroid (17β) dehydrogenase 2
*POLR2A* *	Hs00172187_m1	Polymerase (RNA) II (DNA-directed) polypeptide A, 220 kDa
*RPLP0* *	Hs99999902_m1	Ribosomal protein lateral stalk subunit P0
*STS*	Hs00165853_m1	Steroid sulfatase (microsomal), isozyme S
*SULT1E1*	Hs00193690_m1	Sulfotransferase family 1E, estrogen-preferring, member 1

* Reference genes.

**Table 2 biomolecules-12-00545-t002:** Primer sequences for evaluating gene expression using the SYBR Green assay [33].

Gene Symbol	Gene Name	Forward Primers (5′ to 3′)	Reverse Primers (5′ to 3′)
*ABCC1*	Multidrug-resistance-associated protein 1	GGACTCAGGAGCACACGAAA	ACGGCGATCCCTTGTGAAAT
*ABCC4*	Multidrug-resistance-associated protein 4	AACTGCAACTTTCACGGATG	AATGACTTTTCCCAGGCGTA
*ABCG2*	ATP-binding cassette super-family G member 2	GGGTTTGGAACTGTGGGTAG	AGATGATTCTGACGCACACC
*HPRT1* *	Hypoxanthine-guanine phosphoribosyltransferase	CCTGGCGTCGTGATTAGTC	TGAGGAATAAACACCCTTTCCA
*POLR2A* *	DNA-directed RNA polymerase II subunit RPB1	CAAGTTCAACCAAGCCATTG	GTGGCAGGTTCTCCAAGG
*RPLP0* *	60S acidic ribosomal protein P0	AATGTGGGCTCCAAGCAGAT	TTCTTGCCCATCAGCACCAC
*SLC10A6*	Solute carrier family 10 member 6	TATGACAACCTGTTCCACCG	GAATGGTCAGGCACACAAGG
*SLC22A11*	Solute carrier family 22 member 11	CTCACCTTCATCCTCCCCTG	CCATTGTCCAGCATGTGTGT
*SLC51A*	Organic solute transporter subunit alpha	GCCCTTTCCAATACGCCTTC	TCTGCTGGGTCATAGATGCC
*SLC51B*	Organic solute transporter subunit beta	GTGCTGTCAGTTTTCCTTCCG	TCATGTGTCTGGCTTAGGATGG
*SLCO1A2*	Solute carrier organic anion transporter family member 1A2	GTTGGCATCATTCTGTGCAAATGTT	AACGAGTGTCAGTGGGAGTTATGAT
*SLCO1B1*	Solute carrier organic anion transporter family member 1B1	CAAATTCTCATGTTTTACTG	GATTATTTCCATCATAGGTC
*SLCO1B3*	Solute carrier organic anion transporter family member 1B3	TCCAGTCATTGGCTTTGCAC	TCCAACCCAACGAGAGTCCT
*SLCO1C1*	Solute carrier organic anion transporter family member 1C1	CACACAGACTACCAAACACCC	TCACCATGCCGAACAGAGAA
*SLCO2B1*	Solute carrier organic anion transporter family member 2B1	AGAGCCCTGTGTTCCATTCT	CTCTTGCTCCAGAAATGGCC
*SLCO3A1*	Solute carrier organic anion transporter family member 3A1	CTACGACAATGTGGTCTAC	TTTTGATGTAGCGTTTATAG
*SLCO4A1*	Solute carrier organic anion transporter family member 4A1	ATGCACCAGTTGAAGGACAG	AACAAGGTGGCAGCTTCTGAG
*SLCO4C1*	Solute carrier organic anion transporter family member 4C1	CCAGGAGCCCCAGAAGTC	AACTCGGACAGCGACAGTG

* Reference genes.

**Table 3 biomolecules-12-00545-t003:** Cell lines with defined CC_50_ values for *Cimicifuga racemosa* extract.

Cell Line	CC_50_ (μg/mL)	95% Confidence Interval
HEC-1-A	58.23	40.10–86.40
Ishikawa	23.57	13.95–38.28
RL-95-2	20.16	13.93–28.29
KLE	48.57	31.44–73.59
HIEEC	11.82	6.77–20.31
Kuramochi	63.91	39.12–101.7
COV362	63.48	36.25–109.30
OVSAHO	106.60	66.00–169.50
HIO80	12.56	9.00–17.37

## Data Availability

All data available in Appendix A.

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
