# Peer review of "Physiological Concentrations of Cimicifuga racemosa Extract Do Not Affect Expression of Genes Involved in Estrogen Biosynthesis and Action in Endometrial and Ovarian Cell Lines"

_biomolecules, 2022, doi:10.3390/biom12040545_

Round 1

Reviewer 1 Report

Materials and methods:
1) Сharacterization of the used cell cultures should be expanded: add the grade of malignancy and grade of tumor differentiation for all cultures.
2) Сlearly indicate which concentrations of the test preparation were used for all cell cultures.
3)Explain what the authors mean by indicating "physiological concentrations".

The section Discussion of the results should be presented separately from the Results and expanded by the authors' reasoning.

References:
A more recent literature reference
to describe the molecular subtypes of endometrial cancer should be added. (For example, Talhouk A et all, Cancer 123 (5), 2017)

Author Response

Our response to Reviewers' comments

We thank the reviewers for constructive comments that helped to improve the quality of the manuscript.

Materials and methods:
1) Сharacterization of the used cell cultures should be expanded: add the grade of malignancy and grade of tumor differentiation for all cultures.

We thank the reviewer for this comment, following information has been added:

The HEC-1-A (CVCL_0293) cell line was originally established from moderately differentiated endometrial adenocarcinoma from a 71-year-old patient and was purchased from the American Type Culture Collection (ATCC_HTB-112TM) as p125 on May 31, 2012. McCoy’s 5A Medium (M4892, Sigma-Aldrich GmbH) supplemented with 10% foetal bovine serum (FBS) was used as growth medium. McCoy’s 5A Medium without phenol red (SH30270.01, GE Healthcare Life Sciences) was used as treatment medium. HEC-1-A passage 15 (p+15) cells were authenticated by short tandem repeats (STR) profiling performed by ATCC on February 22, 2018.

The Ishikawa (CVCL_2529) cell line was originally established from a well differentiated endometrial adenocarcinoma from a 39-year-old patient and was purchased from Sigma-Aldrich (ECACC 99040201) as p+3 on December 18, 2012. Minimum Essential Medium Eagle (#M5650) with 2 mM L-glutamine (#G7513) and 5% FBS (#F9665, all from Sigma-Aldrich) was used as growth medium. MEM without phenol red (#51200-038, Thermo Fisher Scientific) and supplemented with 2 mM L-glutamine (#G7153) was used as treatment medium. Ishikawa p+13 cells were authenticated by STR profiling performed by ATCC on February 22, 2018.

The OVSAHO (CVCL_3144) cell line was originally established from a serous papillary adenocarcinoma from a metastatic site in the abdomen (Yanagibashi et al., 1997) of a 56-year-old woman and represent a suitable model of high grade serous OC and was purchased from JCRB (JCRB1046 lot 04062015) as p44 on June 4, 2018. RPMI (R5886) with 10% FBS (F9665) and 2 mM L-glutamine (G7153; all from Sigma-Aldrich GmbH) was used as growth medium. RPMI without phenol red (11835-030; ThermoFisher Scientific) was used as treatment medium.

2) Clearly indicate which concentrations of the test preparation were used for all cell cultures.

Concentrations of CR extract can be found in:

Table S1. Concentrations of Cimicifuga racemosa (CR) extract used for gene expression studies.

Table S2. Concentrations of Cimicifuga racemosa (CR) extract used for Xcelligence experiments in KLE cells.

Furthermore, Table S3 has been added.

Table S3. Concentrations of Cimicifuga racemosa (CR) extract used for CC50 studies.

Cell line

Concentrations of CR extract in μg/mL

HEC-1-A

1, 10, 25, 50, 100

Ishikawa

1, 10, 50, 100, 250, 500

RL-95-2

1, 5, 10, 50, 100, 250, 300

KLE

1, 5, 10, 50, 100, 300, 400

HIEEC

1, 5, 10, 20, 30, 50

Kuramochi

1, 10, 50, 100, 150, 250

COV362

1, 10, 25, 50, 75, 100, 200, 300, 400

OVSAHO

1, 10, 50, 100, 250, 500

HIO-80

1, 5, 10, 25, 50, 100, 150

 3) Explain what the authors mean by indicating "physiological concentrations".

As written in the chapter 3.2 cells were treated with concentrations similar to the plasma concentrations of the 23-epi-26-deoxyactein, a major constituent of CR, measured in women using CR supplements. These were considered as physiological concentrations.

Breemen et al., 2010,( doi: 10.1038/clpt.2009.251) determined that ethanolic CR extract contains 4.375% of 23-epi-26-deoxyactein. After women ingested 32 mg of ethanolic CR extract, around 2 ng/ml peak concentration of 23-epi-26-deoxyactein in plasma were measured. We calculated total CR concentration by dividing 2 ng/ml with 4.375%, giving 45.7 ng/ml and adjusting to CR doses from currently used OTC medication (5 mg per day), giving final concentration of around 7 ng/ml at the plasma peak.

The section Discussion of the results should be presented separately from the Results and expanded by the authors' reasoning.

We kindly thank the reviewer for this comment. Similarly to other recently published papers from Biomolecules that do not have separated Results and Discussion we think that our format is adequate.

https://doi.org/10.3390/biom12040524

https://doi.org/10.3390/biom12040521

https://doi.org/10.3390/biom12040508

 Discussion has been expanded and Figure 8 added as follows:

We have added: “Our results suggest that CR does not influence estrogen  concentration and actions in either normal endometrial cells or well to moderately differentiated endometrial cancer.” to the section 3.2.

To the section 3.4 we have added:

“At 50 μg/ml of CR, SULT1E1 expression was significantly downregulated (Figure 8). This suggests higher flux of estrogens into the cells. If mRNA levels of SULT1E1 correlate to protein levels, less estrone and estradiol is sulphated and therefore can not be removed from the cell even if SLC51B is upregulated. Higher levels of active estrogens could promote proliferation of cells.

Figure 8. Changes in gene expression levels due to high concentrations of Cimicifuga racemosa extract in the HIO-80 cell line (control cell line of ovarian epithelium).”

References:
A more recent literature reference to describe the molecular subtypes of endometrial cancer should be added. (For example, Talhouk A et all, Cancer 123 (5), 2017)

We thank the reviewer for this comment, the reference has been added and text corrected as follows:

Integrated genomic characterisation has further stratified EC cases into four groups: POLE ultramutated, microsatellite instability hypermutated, copy number low and copy number high group [27, 28]. Recently prognostic subgroups have been confirmed, enabling targeted therapy [29].

Reviewer 2 Report

The manuscript was documented for Physiological concentrations of Cimicifuga racemosa extract do 2 not affect expression of genes encoding E1-S transporters, es- 3 trogen biosynthetic and metabolic enzymes in control and cancerous endometrial and ovarian cell lines. The authors should consider reflection of my comments for publication in this journal.

Major comments

1] This manuscript should be revised English and grammatical errors.

2] At the line 264, although the authors estimated cellular viability using trypan blue, the method is not adequate for estimating of the viability. For estimating of early and late apoptosis, authors should use a adequate method such as PI Annexin V

3] At Fig1, authors documented LD50 of CR extract in various cell lines but ordinally, unit of LD50 is w/w. In case of this result, I recommend CC50

4] At Fig1, in all graphs, are the values without error bar not significant ? I couldn't analyse the data. Additionally, if possible, authors should asterisk to all data.

5] At Fig4, are all graphs without a asterisk not significant ? I couldn't analyse the data in fig4

6] Authors should express all data with statistically consistent marking

7] Authors compared efficiency of CR extract among genital cell lines. However, in one cell line, metabolic effects of CR extract were not documented in the results.

Minor comment

1] All P value should be revised P value (italic form)

2] In statistical method, authors should document for used tools such as ANOVA and T test

3] The title should be revised

Author Response

Our response to Reviewers' comments

We thank the reviewer for constructive comments that helped to improve the quality of the manuscript.

1) This manuscript should be revised English and grammatical errors.

Thank you for this comment, since submission the manuscript has been revised and grammatical errors corrected by dr. Eva Lasic.

2) At the line 264, although the authors estimated cellular viability using trypan blue, the method is not adequate for estimating of the viability. For estimating of early and late apoptosis, authors should use a adequate method such as PI Annexin V

We thank the reviewer for this comment. In the literature trypan blue is described as an appropriate reagent for determining cell viability. Moreover, our results were further confirmed with xCelligence RTCA.

doi: 10.1002/0471142735.ima03bs111

https://link.springer.com/protocol/10.1007/978-1-61779-108-6_2

URL: http://tumj.tums.ac.ir/article-1-9559-en.html

3) At Fig1, authors documented LD50 of CR extract in various cell lines but ordinally, unit of LD50 is w/w. In case of this result, I recommend CC50

We thank the reviewer for this comment, we have changed LD50 to CC50.

4) At Fig1, in all graphs, are the values without error bar not significant? I couldn't analyse the data. Additionally, if possible, authors should asterisk to all data.

Please note that all significant differences are marked with asterisk. Raw data has now been provided as Supplementary Material, Tables S4 and S5.

5) At Fig4, are all graphs without a asterisk not significant? I couldn't analyse the data in fig4

Please note that all significant differences are marked with asterisk. Raw data has now been provided as Supplementary Material, Tables S4 and S5.

6) Authors should express all data with statistically consistent marking

We used asterisk sign consistently for all statistically significant different findings.

7) Authors compared efficiency of CR extract among genital cell lines. However, in one cell line, metabolic effects of CR extract were not documented in the results.

We apologize, but we could not identify the missing information.

Minor comment

1) All P value should be revised P value (italic form)

We have corrected this.

2) In statistical method, authors should document for used tools such as ANOVA and T test

We have modified the description to include information about statistical methods used, “Data were statistically analysed Kruskal-Wallis test with Dunn`s multiple comparisons test. P < 0.05 was considered statistically significant.”

3) The title should be revised

The title has been shortened as follows:

Physiological concentrations of Cimicifuga racemosa extract do not affect expression of genes involved in estrogen biosynthesis and action in endometrial and ovarian cell lines

Reviewer 3 Report

In the research article “Physiological concentrations of Cimicifuga racemosa extract do not affect expression of genes encoding E1-S transporters, estrogen biosynthetic and metabolic enzymes in control and cancerous endometrial and ovarian cell lines” Authors using appropriate cell lines and scientific procedures have shown that at physiological levels in patients CR extract most likely does not have estrogenic effect and would probably not affect postmenopausal women’s risk for EC and OC or negatively affect the outcome of EC and OC patients.

Other than the use of ovariectomized mouse model to support their hypothesis they have almost answered all the probable effects of CR on relevant cell lines. I will suggest to reduce the length of the article, at some places specially in the introduction and methodology section there is scope of shortening the manuscript.

Author Response

Our response to Reviewers' comments

We thank the reviewer for constructive comments that helped to improve the quality of the manuscript.

In the research article “Physiological concentrations of Cimicifuga racemosa extract do not affect expression of genes encoding E1-S transporters, estrogen biosynthetic and metabolic enzymes in control and cancerous endometrial and ovarian cell lines” Authors using appropriate cell lines and scientific procedures have shown that at physiological levels in patients CR extract most likely does not have estrogenic effect and would probably not affect postmenopausal women’s risk for EC and OC or negatively affect the outcome of EC and OC patients.

Other than the use of ovariectomized mouse model to support their hypothesis they have almost answered all the probable effects of CR on relevant cell lines. I will suggest to reduce the length of the article, at some places especially in the introduction and methodology section there is scope of shortening the manuscript.

We have shortened the introduction, especially the information regarding OC in introduction:

From:

Among gynaecological cancers, OC is the leading cause of death in the developed world, with most cases diagnosed at stage III or IV of the disease. The prognosis of OC is directly related to the stage of the disease. Similarly as with EC, the incidence of OC increases with age, spiking drastically after the age of 50. OC is a heterogeneous disease and can be categorised into four primary histological subtypes: serous, endometrioid, mucinous, and clear cell. Serous tumours can be further divided into high- and low-grade cancers. High-grade serous cancers represent 70–80% of all OC cases [30, 31]. Higher endogenous estrogen exposure through life increases the risk of OC [31]. Traditionally, epithelial OC was classified into two types. Type 1 is mostly low-grade with well differentiated tumours, characterized by the presence of mutations in KRAS, BRAF, and ERBB2 (which play roles in different cell signalling pathways) and the lack of mutations in TP53. Type 2 usually has mutations in TP53 and abnormalities in BRCA [32]. Recently, high grade serous OC has been divided into four molecular subtypes with distinct genomic profiles: mesenchymal, differentiated, immunoreactive, and proliferative [33].

To:

Among gynaecological cancers, OC is the leading cause of death in the developed world, with most cases diagnosed at stage III or IV of the disease. OC is a heterogeneous disease and can be categorised into four primary histological subtypes: serous, endometrioid, mucinous, and clear cell. High-grade serous cancers represent 70–80% of all OC cases [30, 31]. Higher endogenous estrogen exposure through life increases the risk of OC [31]. Recently, high grade serous OC has been divided into four molecular subtypes with distinct genomic profiles: mesenchymal, differentiated, immunoreactive, and proliferative [32].

And in the methodology section from:

The expressions of genes that encode estrogen receptors and proteins involved in estradiol biosynthesis and oxidative metabolism were examined using quantitative PCR (qPCR). The following was used: exon-spanning hydrolysis probes commercially available as ‘Assay on Demand’ (Applied Biosystems; Foster City, CA, USA) (Table 3), using TaqMan® Fast Advanced Master Mix and universal thermocycling parameters recommended by Applied Biosystems. The expressions of genes that encode for transporters were examined using SYBR Green I Master (Roche, Basel, Switzerland) and primers that were designed in our laboratory (Table 4). Quantification was accomplished with the Applied Biosystemsâ ViiAä 7 Real-Time PCR System (Thermo Fisher Scientific, Waltham, MA, USA). All the cDNA samples were run in triplicates, using 0.25 ml of cDNA, and the reactions were performed in Applied Biosystemsâ MicroAmpâ Optical 384-well plates (Thermo Fisher Scientific, Waltham, MA, USA) in a reaction volume of 5 μL. The PCR amplification efficiency was determined from the slope of the log-linear portion of the calibration curve for each gene investigated, and this was accounted for in further calculations. For gene expression analysis, the normalization factor for each sample was calculated based on the geometric mean of the three most stably expressed reference genes (POLR2A, HPRT1, RPLP0). Gene expression for each sample was calculated from the crossing-point value (Cq) as E−Cq, divided by the normalization factor and multiplied by 1012. The Minimum Information for Publication of Quantitative Real-Time PCR Experiments guidelines were considered in the performance and interpretation of the qPCR reactions [39].

To:

The expressions of genes that encode estrogen receptors and proteins involved in estradiol biosynthesis and oxidative metabolism were examined using quantitative PCR (qPCR). The following was used: exon-spanning hydrolysis probes commercially available as ‘Assay on Demand’ (Applied Biosystems; Foster City, CA, USA) (Table 3), using TaqMan® Fast Advanced Master Mix. The expressions of genes that encode for transporters were examined using SYBR Green I Master (Roche, Basel, Switzerland) and primers that were designed in our laboratory (Table 4). Quantification was accomplished with the Applied Biosystemsâ ViiAä 7 Real-Time PCR System (Thermo Fisher Scientific, Waltham, MA, USA). All the cDNA samples were run in triplicates, using 0.25 ml of cDNA, and the reactions were performed in Applied Biosystemsâ MicroAmpâ Optical 384-well plates (Thermo Fisher Scientific, Waltham, MA, USA) in a reaction volume of 5 μL. For gene expression analysis, the normalization factor for each sample was calculated based on the geometric mean of the three most stably expressed reference genes (POLR2A, HPRT1, RPLP0). Gene expression for each sample was calculated from the crossing-point value (Cq) as E−Cq, divided by the normalization factor and multiplied by 1012. The Minimum Information for Publication of Quantitative Real-Time PCR Experiments guidelines were considered in the performance and interpretation of the qPCR reactions [39].

Round 2

Reviewer 2 Report

No more comments